# GradSkip: Communication-Accelerated Local Gradient Methods with Better Computational Complexity

## Abstract

We study a class of distributed optimization algorithms that aim to alleviate high communication costs by allowing clients to perform multiple local gradient-type training steps prior to communication. While methods of this type have been studied for about a decade, the empirically observed acceleration properties of local training have eluded all attempts at theoretical understanding. In a recent breakthrough, Mishchenko et al. (2022) proved that local training, when properly executed, leads to provable communication acceleration, and this holds in the strongly convex regime without relying on any data similarity assumptions. However, their ProxSkip method requires all clients to take the same number of local training steps in each communication round. Inspired by a common sense intuition, we start our investigation by conjecturing that clients with "less important" data should be able to get away with fewer local training steps without this impacting the overall communication complexity of the method. It turns out that this intuition is correct: we managed to redesign the original ProxSkip method to achieve this. In particular, we prove that our modified method, for which we coined the name GradSkip, converges linearly under the same assumptions and has the same accelerated communication complexity, while the number of local gradient steps can be reduced relative to a local condition number. We further generalize our method by extending the randomness of probabilistic alternations to arbitrary unbiased compression operators and by considering a generic proximable regularizer. This generalization, which we call GradSkip+, recovers several related methods in the literature as special cases. Finally, we present an empirical study on carefully designed toy problems that confirm our theoretical claims.

## 1 Introduction

*Federated Learning (FL)* is an emerging distributed machine learning paradigm where diverse data holders or clients (e.g., smart watches, mobile devices, laptops, hospitals) collectively aim to train a single machine learning model without revealing local data to each other or the orchestrating central server (McMahan et al., 2017; Kairouz et al, 2019; Wang, 2021). Training such models amounts to solving federated optimization problems of the form

$$\min_{x \in \mathbb{R}^d} \left\{ f(x) := \frac{1}{n} \sum_{i=1}^{n} f_i(x) \right\}, \tag{1}$$

where $d$ is the (typically large) number of parameters of the model $x \in \mathbb{R}^d$ we aim to train, and $n$ is the (potentially large) total number of devices in the federated environment. We denote by $f_i(x)$ the loss or risk associated with the data $\mathcal{D}_i$ stored on client $i \in [n] := \{1, 2, \ldots, n\}$. Formally, our goal is to minimize the overall loss/risk denoted by $f(x)$.

Due to their efficiency, *gradient-type methods* with its numerous extensions (Duchi et al., 2011; Zeiler, 2012; Ghadimi and Lan, 2013; Kingma and Ba, 2015; Schmidt et al., 2017; Qian et al., 2019; Gorbunov et al., 2020a) is by far the most dominant method for solving (1) in practice.

The simplest implementation of gradient descent for federated setup requires all workers $i \in [n]$ in each time step $t \geq 0$ to *(i)* compute local gradient $\nabla f_i(x_t)$ at the current model $x_t$, *(ii)* update the current global model $x_t$ using locally computed gradient $\nabla f_i(x_t)$ via (2) with some step size $\gamma > 0$,

*(iii)* average the updated local models $\hat{x}_{i,t+1}$ via (3) to get the new global model $x_{t+1}$.

$$
\begin{aligned}
\hat{x}_{i,t+1} &= x_t - \gamma \nabla f_i(x_t), & (2) \\
x_{t+1} &= \tfrac{1}{n} \sum_{i=1}^{n} \hat{x}_{i,t+1}. & (3)
\end{aligned}
$$

Challenges that characterize FL as a separate distributed training setup, dictating adjustments to the training algorithm, include *high communication costs*, *heterogeneous data distribution*, and *system heterogeneity* across clients. Next, we discuss these challenges and potential algorithmic solutions.

**1.1. Communication Costs.** In federated optimization, communication costs often become a primary bottleneck due to slow and unreliable wireless links between clients and the central server (McMahan et al., 2017). Eliminating the communication step (3) entirely would cause clients to train solely on local data, leading to a poor model because of the limited local data.

A simple trick to reduce communication costs is to perform the costly synchronization step (3) infrequently, allowing multiple local gradient steps (2) in each communication round (Mangasarian, 1995). This trick appears in the celebrated FedAvg algorithm of McMahan et al. (2016; 2017) and its further variations (Haddadpour and Mahdavi, 2019; Li et al., 2019a; Khaled et al., 2019a;b; Karimireddy et al., 2020; Horváth et al., 2022) under the name of *local gradient methods*. However, until very recently, theoretical guarantees on the convergence rates of local gradient methods were worse than the rate of classical gradient descent, which synchronizes after every gradient step.

In a recent line of works (Mishchenko et al., 2022; Malinovsky et al., 2022; Condat and Richtárik, 2022; Sadiev et al., 2022), initiated by Mishchenko et al. (2022), a novel local gradient method, called ProxSkip, was proposed which performs *a random number* of local gradient steps before each communication (alternation between local training and synchronization is probabilistic) and guarantees strong communication acceleration properties. First, they reformulate the problem (1) into an equivalent regularized consensus problem of the form

$$
\min_{x_1,\dots,x_n \in \mathbb{R}^d} \left\{ \tfrac{1}{n} \sum_{i=1}^{n} f_i(x_i) + \psi(x_1,\dots,x_n) \right\}, \quad \psi(x_1,\dots,x_n) := \begin{cases} 0 & \text{if } x_1 = \dots = x_n \\ +\infty & \text{otherwise} \end{cases}, \quad (4)
$$

where communication between the clients and averaging local models $x_1,\dots,x_n$ is encoded as taking the proximal step with respect to $\psi$, i.e., $\mathrm{prox}_{\psi}([x_1 \dots x_n]^{\top}) = [\bar{x} \dots \bar{x}]^{\top}$, where $\bar{x} := \tfrac{1}{n} \sum_{i=1}^{n} x_i$. With this reformulation, ProxSkip method of Mishchenko et al. (2022) performs the proximal (equivalently averaging) step with small probability $p = 1/\sqrt{\kappa}$, where $\kappa$ is the condition number of the problem. Then the key result of the method for smooth and strongly convex setup is $\mathcal{O}(\kappa \log 1/\epsilon)$ iteration complexity with $\mathcal{O}(\sqrt{\kappa} \log 1/\epsilon)$ communication rounds to achieve $\epsilon > 0$ accuracy. Follow-up works extend the method to variance-reduced gradient methods (Malinovsky et al., 2022), randomized application of proximal operator (Condat and Richtárik, 2022), and accelerated primal-dual algorithms (Sadiev et al., 2022). Our work was inspired by the development of this new generation of local gradient methods, also known as Local Training (LT) methods, which we detail shortly.

An orthogonal approach utilizes communication compression strategies on the transferred information. Informally, instead of communicating full precision models infrequently, we might communicate a compressed version of the local model in each iteration via an application of lossy compression operators. Such strategies include sparsification (Alistarh et al., 2018; Mishchenko et al., 2020; Wang et al., 2018), quantization (Alistarh et al., 2017; Sun et al., 2019; Wang et al., 2022), sketching (Hanzely et al., 2018; Safaryan et al., 2021) and low-rank approximation (Vogels et al., 2019).

Our work contributes to the first approach to handling high communication costs that is less understood in theory and, at the same time, immensely popular in the practice of FL.

**1.2. Statistical Heterogeneity.** Because of the decentralized nature of the training data, distributions of local datasets can vary from client to client. This heterogeneity in data distributions poses an additional challenge since allowing multiple local steps would make the local models deviate from each other, an issue widely known as *client drift*. On the other hand, if training datasets are identical across the clients (commonly referred to as a homogeneous setup), then the mentioned drifting issue disappears, and the training can be done without any communication whatsoever. Now, if we interpolate between these two extremes, then under some data similarity conditions (which are typically expressed as gradient similarity conditions), multiple local gradient steps should be useful. In fact, initial theoretical guarantees of local gradient methods utilize such assumptions (Haddadpour and Mahdavi, 2019; Yu et al., 2019; Li et al., 2019b; 2020).

In the fully heterogeneous setup, client drift reduction techniques were designed and analyzed to mitigate the adverse effect of local model deviations (Karimireddy et al., 2020; Gorbunov et al., 2021). A very close analogy is variance reduction techniques called error feedback mechanisms for the compression noise added to lessen the number of bits required to transfer (Condat et al., 2022).

**1.3. System Heterogeneity.** Lastly, system heterogeneity refers to the diversity of clients in terms of their computation capabilities or the amount of resources they are willing to use during the training. In a typical FL setup, all participating clients must perform the same amount of local gradient steps before each communication. Consequently, a highly heterogeneous cluster of devices results in significant and unexpected delays due to slow clients or stragglers.

One approach addressing system heterogeneity or dealing with slow clients is client selection strategies (Luo et al., 2021; Reisizadeh et al., 2020; Wang and Joshi, 2019). Basically, client sampling can be organized in such a way that slow clients do not delay global synchronization, and clients with similar computational capabilities are sampled in each communication round.

Unlike the above strategy, we suggest clients take local steps based on their resources. We consider the full participation setup where each client decides how much local computation to perform before communication. Informally, slow clients do less local work than fast clients, and during the synchronization of locally trained models, the slowdown caused by the stragglers will be minimized.

## 2 Summary of Contributions

We now briefly summarize the key contributions of our work.

**2.1. GradSkip: efficient gradient skipping algorithm.** We design a new local gradient-type method for distributed optimization with communication and computation constraints. The proposed GradSkip (see Algorithm 1) is an extension of the recently developed ProxSkip method (Mishchenko et al., 2022), which was the first method showing communication acceleration property of performing multiple local steps without any data similarity assumptions. GradSkip inherits the same accelerated communication complexity from ProxSkip while further improving computational complexity, allowing clients to terminate their local gradient computations independently from each other.

The key technical novelty of the proposed algorithm is the construction of auxiliary shifts $\hat{h}_{i,t}$ to handle gradient skipping for each client $i \in [n]$. GradSkip also maintains shifts $h_{i,t}$ initially introduced in ProxSkip to handle communication skipping across the clients. We prove that GradSkip converges linearly in strongly convex and smooth setup, has the same $\mathcal{O}(\sqrt{\kappa_{\max}} \log 1/\epsilon)$ accelerated communication complexity as ProxSkip, and requires clients to compute (in expectation) at most $\min(\kappa_i, \sqrt{\kappa_{\max}})$ local gradients in each communication round (see Theorem 3.6), where $\kappa_i$ is the condition number for client $i \in [n]$ and $\kappa_{\max} = \max_i \kappa_i$. Thus, for GradSkip, clients with well-conditioned problems $\kappa_i < \sqrt{\kappa_{\max}}$ perform much less local work to achieve the same convergence rate of ProxSkip, which assumes $\sqrt{\kappa_{\max}}$ local steps on average for all clients.

**2.2. GradSkip+: general GradSkip method.** Next, we generalize the construction and the analysis of GradSkip by extending it in two directions: handling optimization problems with arbitrary proximable regularizer and incorporating general randomization procedures using unbiased compression operators with custom variance bounds. With such enhancements, we propose our second method, GradSkip+ (see Algorithm 2), which recovers several methods in the literature as a special case, including the standard proximal gradient descent (ProxGD), ProxSkip (Mishchenko et al., 2022), RandProx-FB (Condat and Richtárik, 2022) and GradSkip.

**2.3. VR-GradSkip+: reducing the variance of stochastic gradient skipping.** Finally, we propose and analyze variance-reduced extension (see Algorithm 3 in the Appendix) in the case when mini-batch stochastic gradients are implemented instead of full-batch gradients for local computations. Our VR-GradSkip+ method can be viewed as a successful combination of ProxSkip-VR method of Malinovsky et al. (2022) and GradSkip providing computational efficiency through processing smaller batch of samples and probabilistically skipping stochastic gradient computations. We deferred the presentation of the part of our contribution in the appendix due to space limitations.

*Remark* 2.1 (Local Training (LT) vs Accelerated Gradient Descent (AGD)). Nesterov's AGD method Nesterov (2004) matches the communication complexity of our GradSkip algorithm. Its distributed implementation takes one local step per round, suggesting LT methods might lag behind AGD. In

---

**Algorithm 1** GradSkip

---

1: **Input:** stepsize $\gamma > 0$, synchronization probability $p$, probabilities $q_i > 0$ controlling local steps, initial local iterates $x_{1,0} = \cdots = x_{n,0} \in \mathbb{R}^d$, initial shifts $h_{1,0}, \ldots, h_{n,0} \in \mathbb{R}^d$, total number of iterations $T \geq 1$

2: **for** $t = 0, 1, \ldots, T-1$ **do**

3:     **server:** Flip a coin $\theta_t \in \{0,1\}$ with $\mathrm{Prob}(\theta_t = 1) = p$      $\diamond$ Decide when to skip communication

4:     **for all devices $i \in [n]$ in parallel do**

5:         Flip a coin $\eta_{i,t} \in \{0,1\}$ with $\mathrm{Prob}(\eta_{i,t} = 1) = q_i$    $\diamond$ Decide when to skip gradient steps (see Lemma 3.1)

6:         $\hat{h}_{i,t+1} = \eta_{i,t} h_{i,t} + (1 - \eta_{i,t}) \nabla f_i(x_{i,t})$      $\diamond$ Update the local auxiliary shifts $\hat{h}_{i,t}$

7:         $\hat{x}_{i,t+1} = x_{i,t} - \gamma(\nabla f_i(x_{i,t}) - \hat{h}_{i,t+1})$      $\diamond$ Update the local auxiliary iterate $\hat{x}_{i,t}$ via shifted gradient step

8:         **if** $\theta_t = 1$ **then**

9:           $x_{i,t+1} = \frac{1}{n} \sum_{j=1}^{n} \left( \hat{x}_{j,t+1} - \frac{\gamma}{p} \hat{h}_{j,t+1} \right)$   $\diamond$ Average shifted iterates, but only very rarely!

10:         **else**

11:           $x_{i,t+1} = \hat{x}_{i,t+1}$      $\diamond$ Skip communication!

12:         **end if**

13:         $h_{i,t+1} = \hat{h}_{i,t+1} + \frac{p}{\gamma}(x_{i,t+1} - \hat{x}_{i,t+1})$      $\diamond$ Update the local shifts $h_{i,t}$

14:     **end for**

15: **end for**

---

contrast, almost all methods in production are based on local training, as evidenced by FL frameworks like He et al. (2020); Ro et al. (2021); Beutel et al. (2022).

The preference for LT over AGD among practitioners stems from LT's advantages, especially in generalization and communication complexity. Both areas are closely tied with local training, becoming prominent in current research. LT's ability to enhance generalization remains under exploration in FL. Current studies link this improvement to personalization, meta-learning Hanzely and Richtárik (2021); Hanzely et al. (2020), and representation learning Collins et al. (2022). Practically, LT effectively tackles nonconvex challenges, while AGD faces difficulty approximating stationary points of smooth nonconvex functions. Additionally, AGD is more sensitive to the knowledge of the condition number than LT methods, which are versatile and work across a wide range of numbers of local steps.

In statistically heterogeneous cases, AGD often underperforms. Our experiments prove this by showing that when device condition numbers vary, AGD converges slower than GradSkip. Though our work does not primarily aim to directly compare AGD and LT, such a comparative study, to our knowledge, remains a gap in current research and could offer valuable insights.

## 3 GRADSKIP

In this section, we present our first algorithm, GradSkip, and discuss its benefits in detail. Later, we will generalize it, unifying several other methods as special cases. Recall that our target is to address three challenges in FL mentioned in the introductory part, which are *(i)* reduction in communication cost via infrequent synchronization of local models, *(ii)* statistical or data heterogeneity, and *(iii)* reduction in computational cost via limiting local gradient calls based on the local subproblem.

**3.1. Algorithm structure.** For the sake of presentation, we describe the progress of the algorithm using two variables $x_{i,t}, \hat{x}_{i,t}$ for the local models and two variables $h_{i,t}, \hat{h}_{i,t}$ for the local gradient shifts. Essentially, we want to maintain two variables for the local models since clients get synchronized infrequently. The shifts $h_{i,t}$ are designed to reduce the client drift caused by the statistical heterogeneity. Finally, we introduce auxiliary shifts $\hat{h}_{i,t}$ to take care of the different number of local steps. The GradSkip method is formally presented in Algorithm 1.

As an initialization step, we choose probability $p > 0$ to control communication rounds, probabilities $q_i > 0$ for each client $i \in [n]$ to control local gradient steps and initial control variates (or shifts) $h_{i,0} \in \mathbb{R}^d$ to control the client drift. Besides, we fix the stepsize $\gamma > 0$ and assume that all clients

commence with the same local model, namely $x_{1,0} = \cdots = x_{n,0} \in \mathbb{R}^d$. Then, each iteration of the method comprises two stages, the local stage and the communication stage, operating probabilistically. Specifically, the probabilistic nature of these stages is the following. The local stage requires computation only with some predefined probability; otherwise, the stage is void. Similarly, the communication stage requires synchronization between all clients only with probability $p$; otherwise, the stage is void. In the local stage (lines 5–7), all clients $i \in [n]$ in parallel update their local variables $(\hat{x}_{i,t+1}, \hat{h}_{i,t+1})$ using values $(x_{i,t}, h_{i,t})$ from previous iterate either by computing the local gradient $\nabla f_i(x_{i,t})$ or by just copying the previous values. Afterward, in the communication stage (lines 8–13), all clients in parallel update their local variables $(x_{i,t+1}, h_{i,t+1})$ from $(\hat{x}_{i,t+1}, \hat{h}_{i,t+1})$ by either averaging across the clients or copying previous values.

**3.2. Reduced local computation.** Clearly, communication costs are reduced as the averaging step occurs only when $\theta_t = 1$ with probability $p$ of our choice. However, it is not directly apparent how the computational costs are reduced during the local stage. Indeed, both options $\eta_{i,t} = 1$ and $\eta_{i,t} = 0$ involve the expression $\nabla f_i(x_{i,t})$ as if local gradients need to be evaluated in every iteration. As we show in the following lemma, this is not the case.

**Lemma 3.1** (Fake local steps). *Suppose that Algorithm 1 does not communicate for $\tau \geq 1$ consecutive iterates, i.e., $\theta_t = \theta_{t+1} = \cdots = \theta_{t+\tau-1} = 0$ for some fixed $t \geq 0$. Besides, let for some client $i \in [n]$ we have $\eta_{i,t} = 0$. Then, regardless of the coin tosses $\{\eta_{i,t+j}\}_{j=1}^{\tau}$, client $i$ does fake local steps without any gradient computation in $\tau$ iterates. Formally, for all $j = 1, 2, \ldots, \tau + 1$, we have*

$$\hat{x}_{i,t+j} = x_{i,t+j} = x_{i,t}, \quad \hat{h}_{i,t+j} = h_{i,t+j} = h_{i,t} = \nabla f_i(x_{i,t}). \tag{5}$$

Let us reformulate the above lemma. During the local stage of GradSkip, when clients do not communicate with the server, $i^{th}$ client terminates its local gradient steps once the local coin tosses $\eta_{i,t} = 0$. Thus, smaller probability $q_i$ implies sooner coin toss $\eta_{i,t} = 0$ in expectation, hence, less amount of local computation for client $i$. Therefore, we can relax the computational requirements of clients by adjusting these probabilities $q_i$ and controlling the amount of local gradient computations.

Next, let us find out how the expected number of local gradient steps depends on probabilities $p$ and $q_i$. Let $\Theta$ and $H_i$ be random variables representing the number of coin tosses (Bernoulli trials) until the first occurrence of $\theta_t = 1$ and $\eta_{i,t} = 0$ respectively. Equivalently, $\Theta \sim \text{Geo}(p)$ is a geometric random variable with parameter $p$, and $H_i \sim \text{Geo}(1 - q_i)$ are geometric random variables with parameter $1 - q_i$ for $i \in [n]$. Notice that, within one communication round, $i^{th}$ client performs $\min(\Theta, H_i)$ number of local gradient computations, which is again a geometric random variable with parameter $1 - (1 - (1 - q_i))(1 - p) = 1 - q_i(1 - p)$. Therefore, as formalized in the next lemma, the expected number of local gradient steps is $\mathbb{E}[\min(\Theta, H_i)] = 1/(1 - q_i(1 - p))$.

**Lemma 3.2** (Expected number of local steps). *The expected number of local gradient computations in each communication round of GradSkip is $1/(1 - q_i(1 - p))$ for all clients $i \in [n]$.*

Notice that, in the special case of $q_i = 1$ for all $i \in [n]$, GradSkip recovers Scaffnew method of Mishchenko et al. (2022). However, as we will show, we can choose probabilities $q_i$ smaller, reducing computational complexity and obtaining the same convergence rate as Scaffnew.

*Remark* 3.3 (System Heterogeneity). From this discussion, we conclude that GradSkip can also address system or device heterogeneity. In particular, probabilities $\{q_i\}_{i=1}^n$ can be assigned to clients in accordance with their local computational resources; slow clients with scarce compute power should get small $q_i$, while faster clients with rich resources should get bigger $q_i \leq 1$.

**3.3. Convergence theory.** Now that we explained the structure and computational benefits of the algorithm let us proceed to the theoretical guarantees. We consider the same strongly convex and smooth setup as considered by Mishchenko et al. (2022) for the distributed case.

**Assumption 3.4.** All functions $f_i(x)$ are strongly convex with parameter $\mu > 0$ and have Lipschitz continuous gradients with Lipschitz constants $L_i > 0$, i.e., for all $i \in [n]$ and any $x, y \in \mathbb{R}^d$ we have $\frac{\mu}{2}\|x - y\|^2 \leq D_{f_i}(x, y) \leq \frac{L_i}{2}\|x - y\|^2$, where $D_{f_i}(x, y) := f_i(x) - f_i(y) - \langle \nabla f_i(y), x - y \rangle$ is the Bregman divergence associated with $f_i$ at points $x, y \in \mathbb{R}^d$.

We present a Lyapunov-type analysis to prove the convergence, which is a very common approach for iterative algorithms. Consider the Lyapunov function

$$\Psi_t := \sum_{i=1}^n \|x_{i,t} - x_\star\|^2 + \frac{\gamma^2}{p^2} \sum_{i=1}^n \|h_{i,t} - h_{i,\star}\|^2, \tag{6}$$

where $\gamma > 0$ is the stepsize, $x_\star$ is the (necessary) unique minimizer of $f(x)$ and $h_{i,*} = \nabla f_i(x_*)$ is the optimal gradient shift. As we show next, $\Psi_t$ decreases at a linear rate.

**Theorem 3.5.** *Let Assumption 3.4 hold. If the stepsize satisfies* $\gamma \leq \min_i \left\{ \frac{1}{L_i} \frac{p^2}{1-q_i(1-p^2)} \right\}$ *and probabilities are chosen so that* $0 < p$, $q_i \leq 1$, *then the iterates of* GradSkip *(Algorithm 1) satisfy*

$$\mathbb{E}\left[\Psi_t\right] \leq (1-\rho)^t \Psi_0, \quad with \ \rho := \min\{\gamma\mu, 1 - q_{max}(1-p^2)\} > 0. \tag{7}$$

The first and immediate observation from the above result is that, with a proper stepsize choice, GradSkip converges linearly for any choice of probabilities $p$ and $q_i$ from $(0, 1]$. Furthermore, by choosing all probabilities $q_i = 1$ we get the same rate of Scaffnew with $\rho = \min\{\gamma\mu, p^2\}$ (see Theorem 3.6 in (Mishchenko et al., 2022)). If we further choose the largest admissible stepsize $\gamma = 1/L_{\max}$ and the optimal synchronization probability $p = 1/\sqrt{\kappa_{\max}}$, we get $\mathcal{O}(\kappa_{\max} \log 1/\epsilon)$ iteration complexity, $\mathcal{O}(\sqrt{\kappa_{\max}} \log 1/\epsilon)$ accelerated communication complexity with $1/p = \sqrt{\kappa_{\max}}$ expected number of local steps in each communication round. Here, we used notation $\kappa_{\max} = \max_i \kappa_i$ where $\kappa_i = L_i/\mu$ is the condition number for client $i \in [n]$.

Finally, exploiting smaller probabilities $q_i$, we can optimize computational complexity subject to the same communication complexity as Scaffnew. To do that, note that the largest possible stepsize that Theorem 3.5 allows is $\gamma = 1/L_{\max}$ as $\min_i\{\frac{1}{L_i} \frac{p^2}{1-q_i(1-p^2)}\} \leq \min_i \frac{1}{L_i} \leq \frac{1}{L_{\max}}$. Hence, taking into account $\rho \leq \gamma\mu$, the best iteration complexity from the rate (7) is $\mathcal{O}(\kappa_{\max} \log 1/\epsilon)$, which can be obtained by choosing the probabilities appropriately as formalized in the following result.

**Theorem 3.6** (Optimal parameter choices)**.** *Let Assumption 3.4 hold and choose probabilities* $q_i = \frac{1-1/\kappa_i}{1-1/\kappa_{\max}} \leq 1$ *and* $p = 1/\sqrt{\kappa_{\max}}$. *Then, with the largest admissible stepsize* $\gamma = 1/L_{\max}$, GradSkip *enjoys the following properties:*

(i) $\mathcal{O}\left(\kappa_{\max} \log 1/\varepsilon\right)$ *iteration complexity,*

(ii) $\mathcal{O}\left(\sqrt{\kappa_{\max}} \log 1/\varepsilon\right)$ *communication complexity,*

(iii) *for each client* $i \in [n]$, *the expected number of local gradient computations per communication round is*

$$\frac{1}{1-q_i(1-p)} = \frac{\kappa_i(1+\sqrt{\kappa_{\max}})}{\kappa_i + \sqrt{\kappa_{\max}}} \leq \min(\kappa_i, \sqrt{\kappa_{\max}}). \tag{8}$$

This result clearly quantifies the benefits of using smaller probabilities $q_i$. In particular, if the condition number $\kappa_i$ of client $i$ is smaller than $\sqrt{\kappa_{\max}}$, then within each communication round, it does only $\kappa_i$ number of local gradient steps. However, for a client having the maximal condition number (namely, clients $\arg\max_i\{\kappa_i\}$), the number of local gradient steps is $\sqrt{\kappa_{\max}}$, which is the same for Scaffnew. From this, we conclude that, in terms of computational complexity, GradSkip is always better and can be $\mathcal{O}(n)$ times better than Scaffnew (Mishchenko et al., 2022).

## 4 GRADSKIP+

Here, we aim to present a deeper understanding of GradSkip by extending it in two directions and designing our generic GradSkip+ method.

The first direction is the optimization problem's formulation. As we discussed earlier, distributed optimization (1) with consensus constraints can be transformed into a regularized optimization problem (4) in the lifted space. Following Mishchenko et al. (2022), we consider the (lifted) problem[1]

$$\min_{x \in \mathbb{R}^d} f(x) + \psi(x), \tag{9}$$

where $f(x)$ is strongly convex and smooth loss, while $\psi(x)$ is closed, proper and convex regularizer (e.g., see (4)). The requirement we impose on the regularizer is that the proximal operator of $\psi$ is a single-valued function that can be computed.

The second extension in GradSkip+ is the generalization of the randomization procedure of probabilistic alternations in GradSkip by allowing arbitrary unbiased compression operators with certain bounds on the variance. Let us formally define the class of compressors we will be working with.

---

[1]To be precise, the lifted problem is in $\mathbb{R}^{nd}$ as we stack all local variables $x_1, \ldots, x_n \in \mathbb{R}^d$ into one.

---

**Algorithm 2** GradSkip+
---

1: **Parameters:** stepsize $\gamma > 0$, compressors $\mathcal{C}_\omega \in \mathbb{B}^d(\omega)$ and $\mathcal{C}_\Omega \in \mathbb{B}^d(\Omega)$.
2: **Input:** initial iterate $x_0 \in \mathbb{R}^d$, initial control variate $h_0 \in \mathbb{R}^d$, number of iterations $T \geq 1$.
3: **for** $t = 0, 1, \ldots, T-1$ **do**
4:     $\hat{h}_{t+1} = \nabla f(x_t) - (\mathbf{I} + \Omega)^{-1} \mathcal{C}_\Omega (\nabla f(x_t) - h_t)$          $\diamond$ Update the shift $\hat{h}_t$
                   via shifted compression
5:     $\hat{x}_{t+1} = x_t - \gamma(\nabla f(x_t) - \hat{h}_{t+1})$         $\diamond$ Update the iterate $\hat{x}_t$ via shifted gradient step
6:     $\hat{g}_t = \frac{1}{\gamma(1+\omega)}\mathcal{C}_\omega \left(\hat{x}_{t+1} - \mathrm{prox}_{\gamma(1+\omega)\psi}\left(\hat{x}_{t+1} - \gamma(1+\omega)\hat{h}_{t+1}\right)\right)$     $\diamond$ Estimate the proximal
                   gradient
7:     $x_{t+1} = \hat{x}_{t+1} - \gamma\hat{g}_t$             $\diamond$ Update the main iterate $x_t$
8:     $h_{t+1} = \hat{h}_{t+1} + \frac{1}{\gamma(1+\omega)}(x_{t+1} - \hat{x}_{t+1})$             $\diamond$ Update the main shift $h_t$
9: **end for**

---

**Definition 4.1** (Unbiased Compressors). For any positive semidefinite matrix $\Omega \succeq 0$, denote by $\mathbb{B}^d(\Omega)$ the class of (possibly randomized) unbiased compression operators $\mathcal{C} \colon \mathbb{R}^d \to \mathbb{R}^d$ such that for all $x \in \mathbb{R}^d$ we have

$$\mathbb{E}\left[\mathcal{C}(x)\right] = x, \quad \mathbb{E}\left[\|(\mathbf{I} + \Omega)^{-1}\mathcal{C}(x)\|^2\right] \leq \|x\|^2_{(\mathbf{I}+\Omega)^{-1}}.$$

The class $\mathbb{B}^d(\Omega)$ is a generalization of commonly used class $\mathbb{B}^d(\omega)$ of unbiased compressors with variance bound $\mathbb{E}\left[\|\mathcal{C}(x)\|^2\right] \leq (1 + \omega)\|x\|^2$ for some scalar $\omega \geq 0$. Indeed, when the matrix $\Omega = \omega\mathbf{I}$, then $\mathbb{B}^d(\omega\mathbf{I})$ coincides with $\mathbb{B}^d(\omega)$. Furthermore, the following inclusion holds:

**Lemma 4.2.** $\mathbb{B}^d(\Omega) \subseteq \mathbb{B}^d\big((1+\lambda_{\max}(\Omega))^2/(1+\lambda_{\min}(\Omega)) - 1\big)$.

The purpose of this new variance bound with matrix parameter $\Omega$ is to introduce non-uniformity on the compression level across different directions. For example, in the reformulation (4), each client controls $1/n$ portion of the directions and the level of compression. For example, consider compression operator $\mathcal{C} \colon \mathbb{R}^d \to \mathbb{R}^d$ defined as

$$\mathcal{C}(x)_j = \begin{cases} x_j/p_j, & \text{with probability } p_j, \\ 0, & \text{with probability } 1-p_j, \end{cases} \tag{10}$$

for all coordinates $j \in [d]$ and for any $x \in \mathbb{R}^d$, where $p_j \in (0, 1]$ are given probabilities. Then, it is easy to check that $\mathcal{C} \in \mathbb{B}^d(\Omega)$ with diagonal matrix $\Omega = \mathbf{Diag}(1/p_j - 1)$ having diagonal entries $1/p_j - 1 \geq 0$.

With finer control over the compression operator, we can make use of the granular smoothness information of the loss function $f$ via so-called smoothness matrices (Qu and Richtárik, 2016b;a).

**Definition 4.3** (Matrix Smoothness). A differentiable function $f : \mathbb{R}^d \to \mathbb{R}$ is called $\mathbf{L}$-smooth with some symmetric and positive definite matrix $\mathbf{L} \succ 0$ if

$$D_f(x, y) \leq \tfrac{1}{2}\|x - y\|^2_{\mathbf{L}}, \quad \forall x, y \in \mathbb{R}^d. \tag{11}$$

The standard $L$-smoothness condition with scalar $L > 0$ is obtained as a special case of (11) for matrices of the form $\mathbf{L} = L\mathbf{I}$, where $\mathbf{I}$ is the identity matrix. The notion of matrix smoothness provides more information about the function than mere scalar smoothness. In particular, if $f$ is $\mathbf{L}$-smooth, then it is also $\lambda_{\max}(\mathbf{L})$-smooth due to the relation $\mathbf{L} \preceq \lambda_{\max}(\mathbf{L})\mathbf{I}$. Smoothness matrices have been used in the literature of randomized coordinate descent (Richtárik and Takáč, 2016; Hanzely and Richtárik, 2019b;a) and distributed optimization (Safaryan et al., 2021; Wang et al., 2022).

**4.1. Algorithm description.** Similar to GradSkip, we maintain two variables $x_t$, $\hat{x}_t$ for the model, and two variables $h_t$, $\hat{h}_t$ for the gradient shifts in GradSkip+. Initial values $x_0 \in \mathbb{R}^d$ and $h_0 \in \mathbb{R}^d$ can be chosen arbitrarily. In each iteration, GradSkip+ first updates the auxiliary shift $\hat{h}_{t+1}$ using the previous shift $h_t$ and gradient $\nabla f(x_t)$ (line 4). This shift $\hat{h}_{t+1}$ is then used to update the auxiliary iterate $x_t$ via shifted gradient step (line 5). Then we estimate the proximal gradient $\hat{g}_t$ (line 6) in order to update the main iterate $x_{t+1}$ (line 7). Lastly, we complete the iteration by updating the main

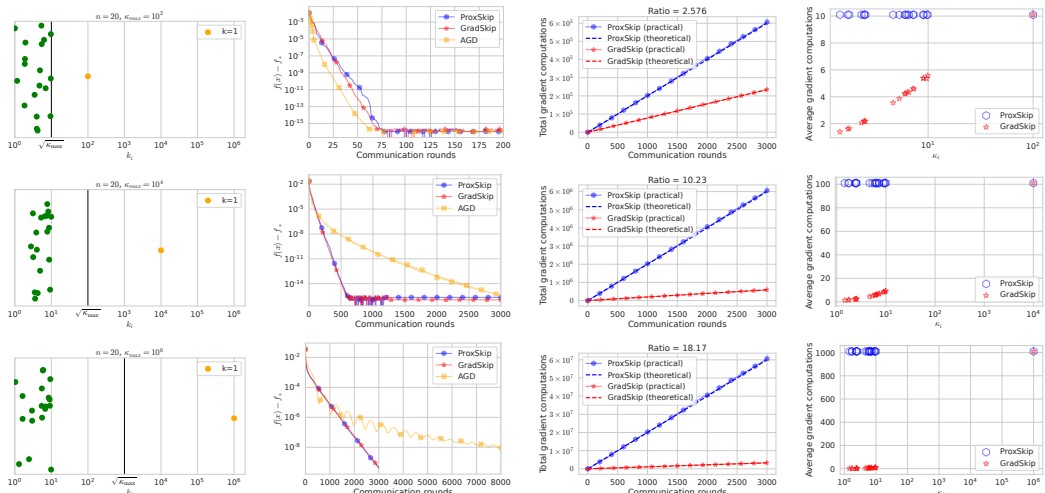

Figure 1: The first column displays the condition numbers for devices. The second column presents convergence per communication round. The third column contrasts theoretical and practical gradient computation counts. The final column reveals the average gradient computations for devices with condition number $\kappa_i$. Notably, in GradSkip, the device with $\kappa_i = \kappa_{max}$ performs gradient computations at a rate comparable to all devices in ProxSkip.

shift $h_t$ (line 8). See Algorithm 2 for the formal steps. In the Appendix D.3, we show that GradSkip+ recovers ProxGD, ProxSkip and RandProx-FB (Condat and Richtárik, 2022) as a special case.

**4.2. Convergence theory.** We now present the convergence theory for GradSkip+, for which we replace the scalar smoothness Assumption 3.4 by matrix smoothness.

**Assumption 4.4** (Convexity and smoothness). We assume that the loss function $f$ is $\mu$-strongly convex with positive $\mu > 0$ and $\mathbf{L}$-smooth with positive definite matrix $\mathbf{L} \succ 0$.

Similar to (6), we analyze GradSkip+ using the Lyapunov function $\Psi_t := \|x_t - x_\star\|^2 + \gamma^2(1 + \omega)^2\|h_t - h_\star\|^2$, where $h_* = \nabla f(x_*)$. The next theorem shows the general linear convergence result.

**Theorem 4.5.** *Let Assumption 4.4 hold, $\mathcal{C}_\omega \in \mathbb{B}^d(\omega)$ and $\mathcal{C}_{\boldsymbol{\Omega}} \in \mathbb{B}^d(\boldsymbol{\Omega})$ be the compression operators, and $\widetilde{\boldsymbol{\Omega}} := \mathbf{I} + \omega(\omega + 2)\boldsymbol{\Omega}(\mathbf{I} + \boldsymbol{\Omega})^{-1}$. Then, if the stepsize $\gamma \leq \lambda_{\max}^{-1}(\mathbf{L}\widetilde{\boldsymbol{\Omega}})$, the iterates of GradSkip+ (Algorithm 2) satisfy*

$$\mathbb{E}\left[\Psi_t\right] \leq (1 - \min\{\gamma\mu, \delta\})^t \Psi_0, \tag{12}$$

*where* $\delta = 1 - \frac{1}{1+\lambda_{\min}(\boldsymbol{\Omega})}\left(1 - \frac{1}{(1+\omega)^2}\right) \in [0, 1]$.

First, if we choose $\mathcal{C}_{\boldsymbol{\Omega}}$ to be the identity compression (i.e., $\boldsymbol{\Omega} = \mathbf{0}$), then GradSkip+ reduces to RandProx-FB and we recover asymptotically the same rate with linear factor $(1 - \min\{\gamma\mu, 1/(1+\omega)^2\})$ (see Theorem 3 of Condat and Richtárik (2022)). If we further choose $\mathcal{C}_\omega$ to be the Bernoulli compression with parameter $p \in (0, 1]$, then $\omega = 1/p - 1$ and we get the rate of ProxSkip.

In order to recover the rate (7) of GradSkip, consider the lifted space $\mathbb{R}^{nd}$ with reformulation (4) and objective function $f(x) = \frac{1}{n}\sum_{i=1}^n f_i(x_i)$, where $x_i \in \mathbb{R}^d$ and $x = (x_1, \ldots, x_n) \in \mathbb{R}^{nd}$. From $\mu$-strong convexity of each loss function $f_i$, we conclude that $f$ is also $\mu$-strongly convex. Regarding the smoothness condition, we have $L_i\mathbf{I} \in \mathbb{R}^{d \times d}$ smoothness matrices (e.g., scalar $L_i$-smoothness) for each $f_i$, which implies that the overall loss function $f$ has $\mathbf{L} = \mathbf{Diag}(L_1\mathbf{I}, \ldots, L_n\mathbf{I}) \in \mathbb{R}^{nd \times nd}$ as a smoothness matrix. Furthermore, choosing Bernoulli compression operators $\mathcal{C}_\omega = \mathcal{C}_p^{nd}$ and $\mathcal{C}_{\boldsymbol{\Omega}} = \mathcal{C}_{q_1}^d \times \cdots \times \mathcal{C}_{q_n}^d$ in the lifted space $\mathbb{R}^{nd}$, we get $\omega = 1/p - 1$ and $\boldsymbol{\Omega} = \mathbf{Diag}(1/q_i - 1)$. It remains to plug all these expressions into Theorem 4.5 and recover Theorem 3.6. Indeed, $\lambda_{\min}(\boldsymbol{\Omega}) = 1/q_{\max} - 1$ and, hence, $\delta = 1 - q_{\max}\left(1 - p^2\right)$. Lastly, Theorem 4.5 recovers the same stepsize bound as $\lambda_{\max}^{-1}(\mathbf{L}\widetilde{\boldsymbol{\Omega}}) = \min_i\left(L_i\left(1 + (1 - q_i)\left(1/p^2 - 1\right)\right)\right)^{-1} = \min_i\left\{\frac{1}{L_i}\frac{p^2}{1-q_i(1-p^2)}\right\}$.

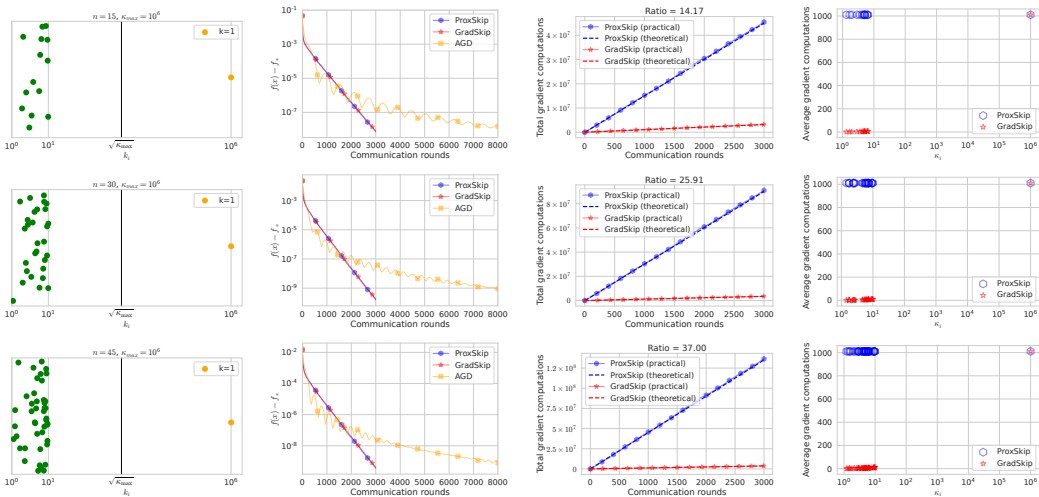

Figure 2: The columns in this figure represent the same as those in Figure 1.

## 5 EXPERIMENTS

To test the performance of GradSkip and illustrate theoretical results, we use the classical logistic regression problem. The loss function for this model has the following form:

$$f(x) = \frac{1}{n} \sum_{i=1}^{n} \frac{1}{m_i} \sum_{j=1}^{m_i} \log \left( 1 + \exp \left( -b_{ij} a_{ij}^\top x \right) \right) + \frac{\lambda}{2} \|x\|^2,$$

where $n$ is the number of clients, $m_i$ is the number of data points per worker, $a_{ij} \in \mathbb{R}^d$ and $b_{ij} \in -1, +1$ are the data samples, and $\lambda$ is the regularization parameter.

We conducted experiments on artificially generated data and on the *"australian"* dataset from LibSVM library (Chang and Lin, 2011) (see Appendix E). All algorithms are implemented in Python using RAY (Moritz et al., 2018) for parallelization. We run all algorithms using their theoretically optimal hyper-parameters (stepsize, probabilities). We compare GradSkip with ProxSkip and AGD, as both have SOTA accelerated communication complexity. However, since AGD doesn't outperform GradSkip in communication complexity, and given the importance of communication complexity in the FL setup, we don't delve into their computational complexities. While ProxSkip-VR has a better computational complexity, the difference in computational complexity between VR-GradSkip+ and ProxSkip-VR is similar to that between GradSkip and ProxSkip, so we also skip comparing them.

The expected number of local gradient computations per communication round for GradSkip is at most $\sum_{i=1}^{n} \min(\kappa_i, \sqrt{\kappa_{\max}})$ (see (8)). In contrast, for ProxSkip, we have $n\sqrt{\kappa_{\max}}$. Therefore, the gradient computation ratio of ProxSkip over GradSkip depends on the number of devices having $\kappa_i \geq \sqrt{\kappa_{\max}}$ condition number. If there are $k \leq n$ such devices, then the gradient computation ratio of ProxSkip over GradSkip converges to $n/k \geq 1$ when $\kappa_{\max} \to \infty$.

In our experiments, only one device has an ill-conditioned local problem ($k = 1$). To showcase this convergence, we generate data to control the smoothness constants and set the regularization parameter $\lambda = 10^{-1} = \mu$. We run GradSkip and ProxSkip algorithms for 3000 communication rounds. Figure 1 features $n = 20$ devices. One device is given a large $L_i = L_{max}$, while the others have $L_i \sim \text{Uniform}(0.1, 1)$. The second column illustrates comparable convergence for GradSkip and ProxSkip. As we increment $L_{\max}$ row by row, the ratio appears to converge to $n = 20$. Conversely, AGD's convergence declines with increasing data heterogeneity, and it only beats GradSkip in the first case by a negligible amount of communication rounds. Figure 2 demonstrates that by increasing the client count ($n$), this ratio can grow significantly. One device is assigned a large $L_i = L_{max} = 10^5$, with the remaining devices set to $L_i \sim \text{Uniform}(0.1, 1)$. As we progress row by row, $n$ increases.

