# OpenReview forum: "GradSkip: Communication-Accelerated Local Gradient Methods with Better Computational Complexity"
_ICLR.cc/2024/Conference — Submitted to ICLR 2024_

### Official Review · Reviewer_su5x · 2023-10-29

**Soundness:** 4 excellent
**Presentation:** 4 excellent
**Contribution:** 3 good
**Rating:** 8
**Confidence:** 4

**Summary:**

This paper addresses an intriguing issue in the domain of distributed optimization algorithms. Conventionally, in these algorithms, clients need to have periodic communication and each client performs an equal number of local training steps per communication round. The authors question this norm, pointing out that some clients might face more complex data or difficult problems, potentially necessitating more local training.

The paper introduces a novel algorithm, GradSkip, which realizes this intuition. The authors also provides a clear mathematical analysis and proof. The paper demonstrates that the number of local gradient steps can be reduced relative to the local condition number without undermining the communication complexity. Furthermore, the paper extends its discussion to include other scenarios like variance reduction and gradient compression, leading to the development of GradSkip+.

**Strengths:**

1. The paper uncovers a notable conclusion that clients with simpler data or problems might require fewer local training steps, a concept not widely addressed in current literature.

2. The authors support their findings with stringent and well-articulated mathematical proofs, enhancing the credibility and academic rigor of their work.

3. The analysis provided is detailed and easy to follow, making the complex concepts accessible to readers.

4. Introduction of unbiased compression operators is a significant technical innovation. This concept broadens the scope for a range of new algorithms, marking a substantial contribution to the field.

5. The paper succeeds in providing a comprehensive framework that not only encompasses many known algorithms (ProxGD, ProxSkip, RandProx-FB) but also suggests the potential for several unknown algorithms through its unbiased compression operator.

**Weaknesses:**

One minor critique is that the paper's theoretical bounds are not tight in constant terms.

**Questions:**

Even though I acknowledges the theoretical contributions of this work, I have a question regarding its practical relevance. Specifically, how severe the issue of statistical heterogeneity is in machine learning? How large is the divergence of curvatures among clients? This question is related to the significance of GradSkip algorithm (and potentially any following works) in real-world scenarios.

---

> ### Author Response · Authors · 2023-11-19
> **Gratitude and Clarification**
>
> Thank you for your positive review of our paper. We greatly appreciate the time and effort you dedicated to providing a thorough evaluation of our work. Your recognition of the strengths of our work is immensely encouraging.
>
> Response to the question:
>
> >Even though I acknowledges the theoretical contributions of this work, I have a question regarding its practical relevance. Specifically, how severe the issue of statistical heterogeneity is in machine learning? How large is the divergence of curvatures among clients? This question is related to the significance of GradSkip algorithm (and potentially any following works) in real-world scenarios.
>
> In Figure 3 in the Appendix (last page), we conducted a similar experiment to that in the main part of the paper, using the ``Australian'' dataset from the LibSVM library. In this experiment with 20 devices, we identified 8 ill-conditioned devices, demonstrating that such highly heterogeneous datasets do exist in real-world scenarios. GradSkip showed almost twice the efficiency compared to ProxSkip in terms of total computational complexity. In the fourth column of the results, it is observed that, on average, 9 out of 20 devices perform 5 times less work than the device with the $L_{\max}$-smoothness constant. Although our experiments on real datasets were limited, we believe that in practical situations where local datasets are highly heterogeneous, significant differences in local smoothness constants $L_i$ are likely. Consider, for instance, a scenario in federated image classification where users from diverse groups contribute their images for model training.

---

> > ### Comment · Reviewer_su5x · 2023-11-23
> > **Reply to author's rebuttal**
> >
> > Indeed, LibSVM is quite limited. I am curious about the potential of GradSkip's core idea, that some clients have simpler tasks and thus can train less, in more complex scenarios. For instance, in federated learning for ResNet image classification, how significant would the variation in optimization difficulty be across different clients?
> >
> > I agree with other reviewers that this work is an extension of ProxSkip. However, I do not see this as a drawback. I think this paper makes unique contributions beyond what ProxSkip offers, advancing the understanding of client-specific training needs in distributed optimization.
> >
> > Overall, I vote "accept" due to the novel idea of varying training intensity across different clients based on the varying task difficulty they encounter.

---

### Official Review · Reviewer_NZhx · 2023-10-31

**Soundness:** 1 poor
**Presentation:** 1 poor
**Contribution:** 1 poor
**Rating:** 5
**Confidence:** 3

**Summary:**

This work proposes a new local gradient-type method for distributed optimization with communication and computation constraints. The proposed method inherits the same accelerated communication complexity from ProxSkip while further improving computational complexity. And two variants of the proposed method, i.e., GradSkip+ and VR-GradSkip+ are proposed.

**Strengths:**

1. A new local gradient-type method for distributed optimization with communication and computation constraints is proposed in this work, which is the extension of the ProxSkip method. The proposed method inherits the same accelerated communication complexity from ProxSkip while further improving computational complexity.

2. And two variants of the proposed method, i.e., GradSkip+ and VR-GradSkip+ are proposed.

**Weaknesses:**

1. The assumption that functions $f_i(x)$ are strongly convex is too strong since many functions will not satisfy this assumption when utilizing neural networks.

2. Lack of theoretical analysis of the communication complexity of the proposed method. In distributed optimization, communication complexity is crucial for minimizing inter-node communication to enhance system efficiency and reduce communication costs.

3. The experimental results are limited, the authors should conduct more experiments to verify the performance of the proposed method.

4. The writing of this work is poor. I can't find the Conclusion section. And the summary of contributions is excessively lengthy.

5. There are lots of mistakes in this work, for example,

``Appendix ??'',

 ``see Algorithm ?? in the Appendix'',

`` (see Appendix)''

**Questions:**

Please see the weakness above.

---

> ### Author Response · Authors · 2023-11-19
> **Rebuttal by Authors**
>
> Thank you for your review. We below address the weaknesses you mentioned.
>
> Weaknesses:
>
> >1. The assumption that functions $f_i(x)$ are strongly convex is too strong since many functions will not satisfy this assumption when utilizing neural networks.
>
> All theory relies on some assumptions: there is no free lunch. Please note that these assumptions are standard in the field. Countless published works use precisely these assumptions. To the best of our knowledge, the phenomena we study exist under these assumptions, and it is not known whether they hold under more relaxed settings. While this is indeed a limitation, it is important to recognize that every theoretical paper has some limitations. The critical factor is whether the limitations are reasonable given the subject being studied. In our work, this is clearly the case. We use the same setup as the ProxSkip work but are able to develop a more refined method with better theoretical properties.
>
> Theory papers can be judged on at least two axes: depth, which refers to the sharpness of the results in a limited setting, and breadth, which involves extensions to other settings. In our work, we primarily focus on depth rather than breadth. We believe that depth is significantly more important - both we and others can work on extensions and generalizations later. However, for such work to be meaningful, it needs to be built on strong or deep foundations.
>
> >2. Lack of theoretical analysis of the communication complexity of the proposed method. In distributed optimization, communication complexity is crucial...
>
> What do you mean by the lack of theoretical analysis of the communication complexity? In Theorem 3.6, we demonstrate the communication complexity of our GradSkip method, which matches that of the state-of-the-art ProxSkip method. The complete proof of this theorem, along with all other statements we make, is included in the appendix.
>
> >3. The experimental results are limited, the authors should conduct more experiments to verify the performance of the proposed method.
>
> Firstly, we'd like to understand your perspective more clearly: what specific types of experiments or settings did you have in mind that would further verify the performance of our proposed method?
>
> The goal of our experiments is to illustrate our theory. In our view, strong theoretical research does not necessarily require extensive experimental validation, just as robust practical work may not always need a theoretical basis to be publishable. Our experiments are carefully designed to clearly illustrate our theoretical claims. We adhere to the principle of Occam's Razor, advocating for simplicity in experimentation when it suffices to validate the theory. This approach, we believe, provides clearer insights than conducting complex or excessive experiments.
>
> Our experiments focus on comparing our algorithm, GradSkip, with ProxSkip – the only known local training algorithm with accelerated communication complexity available at the time of our research. We chose not to compare with algorithms that lack accelerated communication complexity since our primary goal was to enhance computational complexity within the realm of local training algorithms with accelerated communication. The ProxSkip research already demonstrates that previous generations of local training methods fall short in terms of communication complexity. Thus, our comparison with the state-of-the-art (SOTA), ProxSkip, is both relevant and sufficient for the scope of our study.
>
> In light of this, do you agree that our experiments sufficiently and effectively corroborate our theoretical claims?
>
> >4. The writing of this work is poor. I can't find the Conclusion section. And the summary of contributions is excessively lengthy.
>
> We understand your concerns regarding the Conclusion section and the summary of contributions.
>
> Due to space constraints, we have integrated the key points typically found in a Conclusion section within other parts of the paper. We believe this approach still effectively communicates the core findings and implications of our research. However, for the camera-ready version of the paper, we will review it again to ensure that these key points are clearly highlighted and easily identifiable to the reader.
>
> Regarding the summary of contributions, we have strived to be as concise as possible while ensuring that all critical aspects of our work are adequately covered. We will take another look at this section to see if any further streamlining is possible without omitting essential details.
>
> >5. There are lots of mistakes in this work...
>
> We apologize for the LaTeX errors in our initial submission. These were formatting issues, and we assure you they have been corrected. In the full version of the paper in the supplementary materials, these issues do not exist. Furthermore, we have recently uploaded a revised version of the main part of the paper, where all these issues have been addressed and fixed.

---

> ### Comment · Reviewer_NZhx · 2023-11-22
>
> Thanks for your responses. My concerns have not been resolved. It's quite common for theoretical proofs to include certain assumptions, but these assumptions can be categorized as either strong or mild. In your case, the assumption that f needs to simultaneously satisfy both L-smooth and u-strongly convex is not a very mild assumption in the field of Federated Learning [1, 2, 3, 4], especially when you're using neural networks.
>
> [1] Local SGD with Periodic Averaging: Tighter Analysis and Adaptive Synchronization
>
> [2] On the Convergence of Communication-Efficient Local SGD for Federated Learning
>
> [3] Faster non-convex federated learning via global and local momentum
>
> [4] Asynchronous Parallel Stochastic Gradient for Nonconvex Optimization
>
> So, it's only then that I would consider the scenarios where the proposed work can be applied are limited, and that's why I requested you to supplement more experiments to demonstrate the generality of the proposed work. However, I haven't seen a satisfactory response.

---

### Official Review · Reviewer_8o1X · 2023-11-03

**Soundness:** 2 fair
**Presentation:** 3 good
**Contribution:** 2 fair
**Rating:** 5
**Confidence:** 4

**Summary:**

The paper proposes Gradskip for solving federated optimization problems with smooth strongly convex objective. Gradskip improves local gradient computation complexity and achieves the optimal communication complexity. The paper further extends the idea of Gradskip to propose Gradskip+ and VR-Gradskip+, which covers a wider range of application.

**Strengths:**

1. The proposed Gradskip method and its extensions modify Scaffnew by allowing skipping local gradient computation and improve the local gradient computation complexity to $O(\min(\sqrt{\kappa_{\max}},\kappa_i)\log(1/\epsilon))$ from $O(\sqrt{\kappa_{\max}}\log(1/\epsilon))$, while still achieving the optimal communication complexity $\sqrt{\kappa}\log(1/\epsilon)$. I suggest the authors summarize their results and existing work in table.
2. Allowing skipping gradient computation is helpful to address system heterogeneity as slow clients can compute less in a communication round.

**Weaknesses:**

1. The novelty of this paper looks somewhat limited. The novelty and main contribution is that Gradskip doesn't always compute local gradient and thus requires $O(\min(\sqrt{\kappa_{\max}},\kappa_i)\log(1/\epsilon))$ proposes Gradskip, instead of $O(\sqrt{\kappa_{\max}}\log(1/\epsilon))$. However, the framework and analysis of proposed Gradskip is similar to Scaffnew.
2. The improvement on computational cost heavily depends on the values of $q_i$, which rely on $\kappa_i$. However, Remark 3.3 says GradSkip addresses heterogeneity by assigning $q_i$ to clients in accordance with their local computational resources. It is unclear how to connect $\kappa_i$ to the local computational resources.
3. Can Gradskip also make improvement on computation time over Scaffnew? What is the time cost for computing gradient in each iteration?

**Questions:**

see the section of weaknesses

---

> ### Author Response · Authors · 2023-11-19
> **Rebuttal by Authors 1**
>
> Thank you for reviewing our work and taking the time to look at our submission. We're glad to have the chance to respond to your comments and questions. Here are our answers to the points you brought up. If you have any more questions or concerns, please let us know.
>
> Addressing Weaknesses:
>
> >1. The novelty of this paper looks somewhat limited. The novelty and main contribution is that Gradskip doesn't always compute local gradient and thus requires $O(\min(\sqrt{\kappa_{max}}, \kappa_i) \log(1/\epsilon))$. However, the framework and analysis of proposed Gradskip is similar to Scaffnew.
>
> The novelty in GradSkip is an extension of the original ProxSkip method in various directions. We provide the entire Section 2 to summarize our contributions.
>
> We manage to modify ProxSkip in such a way that we preserve its accelerated communication complexity while provably improving its computation complexity. That is, our local training method, GradSkip, is still fast from a communication round viewpoint but "bothers" each client only as much as necessary, depending on the quality/importance of the local data stored by each client. This is the first time a result of this type appears in the local training / federated learning literature. In all prior works, there is no theoretical result that would allow the clients to take a different number of local steps and, at the same time, benefit from this. This is the main and key contribution of our work.
>
> Starting from the second paragraph of Section 2.1, we discuss the key technical innovations behind GradSkip, primarily the construction of auxiliary shifts. We believe that the design and provision of rigorous convergence guarantees for auxiliary shifts, particularly those tailored to address specific constraints, is a complex and nuanced task requiring thorough consideration. Additionally, we believe that a small change leading to a significant impact is often more surprising and valuable than a larger change achieving the same outcome.
>
> >2. The improvement on computational cost heavily depends on the values of $q_i$, which rely on $\kappa_i$. However, Remark 3.3 says GradSkip addresses heterogeneity by assigning $q_i$ to clients in accordance with their local computational resources. It is unclear how to connect $\kappa_i$ to the local computational resources.
>
> Our selection of optimal values for $q_i$, which depend on the condition numbers $\kappa_i$, leads to a reduction in the total number of gradient computations while keeping the communication complexity unchanged. However, it's crucial in federated learning to minimize unnecessary client involvement, i.e., to use their devices for local training as infrequently as possible. This consideration is somewhat independent of time complexity. Clearly, if there are two training methods that take the same amount of time, but one requires less client engagement, then this approach is preferable, all other factors being equal.
>
> An alternative strategy involves considering the computational power of the devices when determining the $q_i$ values. This approach can significantly reduce training time, as we discussed in the 'System Heterogeneity' subsection of the Introduction. Essentially, slower clients are assigned lower $q_i$ values to lessen their local workload, whereas faster clients receive higher $q_i$ values. This method aims to minimize delays caused by slower clients during model synchronization. For a clearer understanding of this setup and the choice of $q_i$ values, let us provide a more detailed explanation.
>
> Let $T_i$ denote the time required for one local step on client $i$, where $i=1,2,\ldots,n$. We assume that these $T_i$ values can be measured on each device $i$. Then, the average time required to perform local training before communication on client $i$ is given by:
> $$
> \frac{T_i}{1-q_i(1-p)},
> $$
> since on average device $i$ does $\frac{1}{1-q_i(1-p)}$ local steps (see Lemma 3.2). We can minimize the waiting time by choosing appropriate values of $q_i$ for each device $i$. We start by setting $q_i=1$ for the fastest clients, i.e., for clients $i\in\arg\min_i(T_i)$. For the remaining clients, we choose $q_i$ such that the average time taken for local training before communication is equal to the average time taken by the fastest device. Mathematically, this can be expressed as
> $$
> \frac{T_i}{1-q_i(1-p)} = \frac{T_{min}}{p},
> $$
> yielding the value of $q_i$ as
> $$
> q_i = \max \left \\{ \frac{1-p\frac{T_i}{T_{min}}}{1-p}, 0 \right \\}.
> $$
>
> Using these values of $q_i$, we can minimize the time delay during communication. It is worth noting that in practice, we can update the values of $q_i$ during training if the estimates of $T_i$ change.
>
> We have not elaborated on this in great detail in the main part of the paper, but we are ready to include a more comprehensive explanation in the appendix of the camera-ready version.

---

> > ### Author Response · Authors · 2023-11-19
> > **Rebuttal by Authors 2**
> >
> > >3. Can Gradskip also make improvement on computation time over Scaffnew? What is the time cost for computing gradient in each iteration?
> >
> > With the optimal choices of $q_i$ as outlined in Theorem 3.6, GradSkip can never be slower than Scaffnew. This is because all devices will never perform more local steps than they would in Scaffnew. That being said, the difference in computation time will depend on the relationship between $\kappa_i$ and the power of device $i$. If the slowest device has $\kappa_{\max}$, both GradSkip and Scaffnew will take the same time in each round, as they will wait for the slowest device, which takes the maximum iterations. However, if the slowest device has $\kappa_i < \sqrt{\kappa_{\max}}$, in the case of GradSkip, it will perform fewer local steps than in Scaffnew, thereby leading to better time complexity than Scaffnew. Furthermore, if the goal is to minimize time complexity, we can choose $q_i$ values considering the devices' time complexity, as we explained earlier in the answer to question 2.

---

### Official Review · Reviewer_ENdC · 2023-11-08

**Soundness:** 3 good
**Presentation:** 3 good
**Contribution:** 2 fair
**Rating:** 5
**Confidence:** 4

**Summary:**

Built upon ProxSkip, authors proposed GradSkip (and variants GradSkip+) algorithms by incorporating new randomness with each client. The proposed algorithm attains better computation complexity compared to existing works.

**Strengths:**

1. The key novelty lies in the newly introduced client-wise randomness, which induces fake local steps and less local steps (Lemma 3.1 and 3.2), the idea is elegant.
2. Better computation complexity.

**Weaknesses:**

1. Compared to ProxSkip (Mishchenko et al. (2022)), the algorithm here requires finer structure information from the devices, i.e., individualized function smoothness parameters, while ProxSkip only requires a global smoothness parameter. And all clients are required to coordinate in advance to know the global information $\kappa_{\max}$, which may be a bit unrealistic.
2. According to Theorem 3.6, the client gradient query number is improved from $\sqrt{\kappa_{\max}}$ to $\min(\kappa_i, \sqrt{\kappa_{\max}})$, while the iteration and communication complexity does not change, the claimed $O(n)$ superiority only appears in scenarios where the devices are very unbalanced (most of them have small $\kappa$, while few of them attain very large $\kappa$. As mentioned in your experiments, only one ill-conditioned device), I may view such scenarios to be relatively rare in real world (or it is better if authors can rationalize it). If so the derived improvement seems to be a little bit weak.
3. As far as I understand, the proof heavily relies on the proof of ProxSkip, which restricts the significance of the contribution a bit.

To summarize, I think the algorithm is an interesting extension of ProxSkip with an elegant modification, while I concern that the improvement may be a bit marginal to cross the bar. Please definitely indicate if I misunderstood any points. Thank you very much for your efforts.

**Questions:**

1. In Assumption 3.4, why not extend each $f_i$ to attain a personalized strong convexity parameter $\mu_i$? I think it should be expected.
2. As a separate question, compared to communication complexity, whether improving individual computation complexity is an important question to the FL community, I expect that such improvement should be attractive to marginalized devices.

---

> ### Author Response · Authors · 2023-11-19
> **Rebuttal by Authors 1: Addressing Weaknesses**
>
> Thank you for your review and the time you invested in evaluating our submission. We appreciate the opportunity to address the points you raised. Here are our responses to the weaknesses and questions you highlighted. Please reply if you have any remaining questions or concerns.
>
> Addressing Weaknesses:
>
> > 1. Compared to ProxSkip (Mishchenko et al. (2022)), the algorithm here requires finer structure information from the devices, i.e., individualized function smoothness parameters, while ProxSkip only requires a global smoothness parameter. And all clients are required to coordinate in advance to know the global information $\kappa_{max}$, which may be a bit unrealistic.
>
> When ProxSkip is applied to Federated Learning, it reduces to Scaffnew. In this case, it requires that each $f_i$ be $L$-smooth and $\mu$-strongly convex (see **Assumption 4.1** in [1]). Therefore, in a heterogeneous setting where each $f_i$ is $L_i$-smooth, this assumption is satisfied with $L = L_{\max}$. In the case of ProxSkip, finding $L = L_{\max}$ involves determining the $L_i$-smoothness constants for each device and then taking their maximum. GradSkip also requires finding the $L_i$ constants, but it benefits from the fact that each $f_i$ has a different $L_i$, leading to less work compared to ProxSkip. Thus, the task of determining $L_i$-smoothness constants remains relevant for ProxSkip as well. Although we did not specifically focus on this aspect in our work, it is worth noting that $L_i$-smoothness can be practically estimated using backtracking line-search strategies (see [2], [3], [4]). Our primary aim was to show that computational complexity can be significantly reduced by exploiting the heterogeneity of devices.
>
> > 2. According to Theorem 3.6, the client gradient query number is improved from $\sqrt{\kappa_{max}}$ to $min(\kappa_i, \sqrt{\kappa_{max}})$, while the iteration and communication complexity does not change, the claimed $O(n)$ superiority only appears in scenarios where the devices are very unbalanced (most of them have small $\kappa$, while few of them attain very large $\kappa$). As mentioned in your experiments, only one ill-conditioned device), I may view such scenarios to be relatively rare in real world (or it is better if authors can rationalize it). If so the derived improvement seems to be a little bit weak.
>
> In Figure 3 in the Appendix (last page), we conducted a similar experiment to that in the main part of the paper, using the ``Australian'' dataset from the LibSVM library. In this experiment with 20 devices, we identified 8 ill-conditioned devices, demonstrating that such highly heterogeneous datasets do exist in real-world scenarios. GradSkip showed almost twice the efficiency compared to ProxSkip in terms of total computational complexity. In the fourth column of the results, it is observed that, on average, 9 out of 20 devices perform 5 times less work than the device with the $L_{\max}$-smoothness constant. Although our experiments on real datasets were limited, we believe that in practical situations where local datasets are highly heterogeneous, significant differences in local smoothness constants $L_i$ are likely. Consider, for instance, a scenario in federated image classification where users from diverse groups contribute their images for model training.
>
> >3. As far as I understand, the proof heavily relies on the proof of ProxSkip, which restricts the significance of the contribution a bit.
>
> Since GradSkip is a generalization of ProxSkip (Scaffnew concretely) by adding an additional randomness, it does rely on the proof of the ProxSkip. But we think that it does not restrict the significance of our contribution. With such a small trick, we can get such a huge advantage over ProxSkip. We think that a small change leading to a large effect is much more surprising and valuable than a large change achieving the same.
>
>
> [1] Mishchenko, K., Malinovsky, G., Stich, S., Richtarik, P.. (2022). ProxSkip: Yes! Local Gradient Steps Provably Lead to Communication Acceleration! Finally!. Proceedings of the 39th International Conference on Machine Learning, in Proceedings of Machine Learning Research 162:15750-15769 Available from https://proceedings.mlr.press/v162/mishchenko22b.html.
>
> [2] Goldstein, A. A. (1962). Cauchy's method of minimization. Numerische Mathematik 4 (1): 146-150.
>
> [3] Armijo, L. (1966). Minimization of functions having Lipschitz continuous first partial derivatives. Pacific Journal of mathematics, 16 (1), 1-3.
>
> [4] Y. Nesterov. (1983). A method of solving a convex programming problem with convergence rate O(1/k2 ). Soviet Mathematics Doklady, 27(2):372–376.

---

> > ### Author Response · Authors · 2023-11-19
> > **Rebuttal by Authors 2: Addressing Questions**
> >
> > Questions:
> >
> > >1. In Assumption 3.4, why not extend each $f_i$ to attain a personalized strong convexity parameter $\mu_i$? I think it should be expected.
> >
> > Yes, it is possible to extend to the case where each device has a different strong convexity parameter $\mu_i$. We didn't include it because the heterogeneity in the $\mu_i$ strong convexity parameters doesn't contribute to the number of local steps, i.e., to the choice of optimal $q_i$. Here's why:
> >
> > First, **Theorem 3.5** will change in the following way if we consider different $\mu_i$s:
> >
> > ***Theorem 3.5.**
> > Let each $f_i$ be $L_i$-smooth and $\mu_i$-strongly convex. If the stepsize satisfies $\gamma \le \min_i\left\\{\frac{1}{L_i}\frac{p^2}{1-q_i\left(1-p^2\right)}\right\\}$ and probabilities are chosen so that $0<p,\, q_i \le 1$, then the iterates of GradSkip (Algorithm 1) satisfy
> > $$
> > E[\Psi_t] \le (1 - \rho)^t \Psi_0,
> >     \quad \text{with } \rho := \min\\{\gamma\mu_{\min},1 - q_{\max}(1-p^2)\\} > 0.
> > $$*
> >
> >
> > The change here is only in $\rho$ where we have $\mu_{\min}$ instead of $\mu$.
> >
> > In this case, the optimal $p$ becomes $p^2=\frac{\mu_{\min}}{L_{\max}}$, as we aim to choose $p$ as small as possible since $p$ is the probability when communication occurs, and we prioritize communication over computation. Following the optimal parameter choices as in **Theorem 3.6**, we should choose $q_i = \frac{1-\frac{\mu_{\min}}{L_i}}{1-\frac{\mu_{\min}}{L_{\max}}}$ so it satisfies $0\le q_i\le 1$ and $q_{\max}=1$. Here, we observe that it doesn't depend on $\mu_i$, thereby indicating that the communication complexity doesn't depend on the heterogeneity of $\mu_i$.
> >
> >
> > >2. As a separate question, compared to communication complexity, whether improving individual computation complexity is an important question to the FL community, I expect that such improvement should be attractive to marginalized devices.
> >
> > What do you mean by marginalized devices? If by 'marginalized devices' you mean those within the network with lower computational power or resources, then certainly, reducing computational complexity is crucial. This aspect is particularly vital for ensuring that these less powerful devices can contribute effectively to the Federated Learning (FL) process without being limited by their capabilities. This concept is addressed as 'System Heterogeneity' in Section 1.3 of our work.
> >
> > However, the significance of reducing computational complexity goes beyond merely assisting marginalized devices. It's beneficial universally, regardless of a device's computational power. In data center scenarios, for instance, lowering computational demands enables devices to dedicate resources to other tasks, thereby enhancing overall efficiency. Similarly, in Federated Learning (FL) scenarios, such as when local training occurs on mobile phones, these devices are not constantly available for computation. Reducing the computational burden is essential to ensure that the FL process is compatible with the primary functions of the mobile phones, preventing resource overload.
> >
> > To summarize, optimizing computation complexity is a fundamental concern in FL. It is advantageous not only for less powerful devices but also for maximizing efficiency across the entire spectrum of devices in the network.

---

> > > ### Comment · Reviewer_ENdC · 2023-11-23
> > > **Thank you**
> > >
> > > Thanks authors for the detailed response.
> > >
> > > I agree that the resemblance with ProxSkip in terms of technical novelty should not be a main reason for rejection. But I still concern that the main claimed superiority of the proposed algorithm only stands out at a (potentially) relative corner case. So I keep my score here as to reflect my evaluation in terms of the significance. To further enhance the significance, maybe authors can further extend the study into the strictly convex case, or turn to show the optimality of the algorithm (lower bound) with the finer local Lipschitz constant structure. Thank you for the efforts.

---

### Meta-Review · Area_Chair_SoSz · 2023-12-12

**Metareview:**

The main contribution of this work lies in reducing the number of local steps taken during a local training approach to federated learning.

The recent Scaffnew algorithm established for the first time that, for strongly convex and smooth federated optimization problems, local training can converge as fast as gradient descent in the heteregenous data regime, with fewer rounds of communication than GD. This approach still required each worker to take the same number of steps of local training.

This work proposes an algorithm that has the same communication complexity and linear convergence as Scaffnew, while allowing the n workers to adjust the amount of local steps based on their local condition number. It is thus possible for the proposed algorithm to have total computational complexity up to O(n) smaller than that of Scaffnew.

The key technical change in the algorithm that enables this reduction is the use of per-worker randomization to control the average number of gradient calls that the workers take between communication. The claimed reduction in local work is achieved when these probabilities are chosen using knowledge of the per-worker condition number.

A general concern shared by most reviewers is that the improvement, both in terms of the analysis and the practical impact, of the algorithm over Scaffnew is marginal.

**Justification For Why Not Higher Score:**

The reviewers mostly concur that the contribution is not sufficiently impactful.

**Justification For Why Not Lower Score:**

N/A

---

### Decision · Program_Chairs · 2024-01-16

Reject